# Toward an Automatic Assessment of Cognitive Dysfunction in Relapsing–Remitting Multiple Sclerosis Patients Using Eye Movement Analysis

**DOI:** 10.3390/s22218220

**Published:** 2022-10-27

**Authors:** Cecilia E. García Cena, David Gómez-Andrés, Irene Pulido-Valdeolivas, Victoria Galán Sánchez-Seco, Angela Domingo-Santos, Sara Moreno-García, Julián Benito-León

**Affiliations:** 1Escuela Técnica Superior de Ingeniería y Diseño Industrial, Centre for Automation and Robotics, ETSIDI-CAR, Universidad Politécnica de Madrid, 28012 Madrid, Spain; 2Child Neurology Unit, Hospital Universitari Vall d’Hebron, Vall d’Hebron Research Institute (VHIR), Euro-NMD and ERN-RND, 08035 Barcelona, Spain; 3Anatomy, Histology and Neuroscience Department, Universidad Autónoma de Madrid, 28049 Madrid, Spain; 4Department of Neurology, University Hospital “Virgen de la Salud”, 45004 Toledo, Spain; 5Department of Neurology, “La Mancha Centro” General Hospital, Alcázar de San Juan, 13600 Ciudad Real, Spain; 6Department of Neurology, University Hospital “12 de Octubre”, 28041 Madrid, Spain; 7Centro de Investigación Biomédica en Red sobre Enfermedades Neurodegenerativas (CIBERNED), 28029 Madrid, Spain; 8Department of Medicine, Complutense University, 28040 Madrid, Spain

**Keywords:** multiple sclerosis, cognitive dysfunction, ocular markers, statistical analysis, prediction methods, video-oculography, eye movements

## Abstract

Despite the importance of cognitive function in multiple sclerosis, it is poorly represented in the Expanded Disability Status Scale (EDSS), the commonly used clinical measure to assess disability, suggesting that an analysis of eye movement, which is generated by an extensive and well-coordinated functional network that is engaged in cognitive function, could have the potential to extend and complement this more conventional measure. We aimed to measure the eye movement of a case series of MS patients with relapsing–remitting MS to assess their cognitive status using a conventional gaze tracker. A total of 41 relapsing–remitting MS patients and 43 age-matched healthy controls were recruited for this study. Overall, we could not find a clear common pattern in the eye motor abnormalities. Vertical eye movement was more impaired in MS patients than horizontal movement. Increased latencies were found in the prosaccades and reflexive saccades of antisaccade tests. The smooth pursuit was impaired with more corrections (backup and catchup movements, p<0.01). No correlation was found between eye movement variables and EDSS or disease duration. Despite significant alterations in the behavior of the eye movements in MS patients, which are compatible with altered cognitive status, there is no common pattern of these alterations. We interpret this as a consequence of the patchy, heterogeneous distribution of white matter involvement in MS that provokes multiple combinations of impairment at different points in the different networks involved in eye motor control. Further studies are therefore required.

## 1. Introduction

Multiple sclerosis (MS) is a chronic inflammatory demyelinating disease with severe implications for its sufferers’ health-related quality of life [1,2]. Patients with MS and their families must adjust to new lifestyle changes and restrictions due to the wide range of signs and symptoms [3,4]. Eye movement abnormalities are frequent among the plethora of motor and non-motor features of MS [5]. These abnormalities can be categorized into two. The first comprises gross eye abnormalities discovered during a routine neurological examination; these include loss of visual acuity and reduction in visual fields, ocular misalignment, different types of abnormal nystagmus, impaired vestibular–ocular response, and ocular motor palsy [5]. These anomalies may result from direct injuries in the cranial nerves, brainstem, and cerebellar circuits that directly regulate eye movement [5]. The second is the possibility of MS subtly interfering with the complex network that supervises eye movement and involves multiple cortical areas and subcortical structures [5]. MS can disrupt the synchronization of neural circuits that connect ocular centers in frontal areas such as the frontal eye fields, supplementary frontal eye fields, dorsolateral prefrontal cortex, and anterior cingulate cortex to those in the posterior parietal cortex [5]. Two examples of this in MS are the dysfunctional attentional control of eye movement in response to distractions [6] and impaired attention interfering with memory-guided saccades [7]. The assessment of eye movements may be a powerful tool for analyzing cognitive changes in MS, acting as a surrogate measure of performance that complements more traditional measures.

In the lives of patients with MS, cognitive dysfunction throws a long shadow. The prevalence of cognitive dysfunction in MS ranges from 40% to 70% [8,9]. The most commonly affected cognitive domains are learning, memory, attention, information processing speed, visuospatial abilities, and executive function [8,9]. Cognitively impaired MS patients are more likely to be unemployed and have constraints in social activities and family responsibilities than patients with equal levels of physical disability [10]. The most commonly validated neuropsychological batteries used to assess cognitive dysfunction in MS are Rao’s Brief Repeatable Battery of Neuropsychological Tests [11] and the Minimal Assessment of Cognitive Function in Multiple Sclerosis [12]. These batteries are complex, time consuming, and require trained specialists to perform the test. Therefore, the assessment of the cognitive impairment in these patients is not extended to clinical settings. Furthermore, despite the importance of cognitive function in MS, it is poorly represented in the Expanded Disability Status Scale (EDSS) [13], which is the commonly used clinical measure to assess disability, suggesting that an analysis of eye movement, which is generated by an extensive and well-coordinated functional network that is engaged in cognitive function, could have the potential to extend and complement this more conventional measure.

Over the last ten years, the measurement of eye movement using video-oculography has increased due to the accuracy of the gaze position computed from image processing [14]. However, in the case of MS, there is no consensus on which visual tests should be applied. Previous analyses of eye movement in MS have not been completed automatically, i.e., with standard paradigms easy to apply (not only for research purposes) that diminished the required degree of expertise [15,16].

We aimed to measure the eye movement of a case series of MS patients with relapsing–remitting MS to assess their cognitive status and find a pattern that allows reaching a consensus on which eye visual tests should be completed to automatize the process. We compared the findings with those obtained from healthy volunteers in the same range of age. It would be a step toward the automatic analysis of the eye movements in MS needed in a clinical setting.

This article is organized as follows. Section 2 describes the materials used and methods followed to accomplish this study. Section 3 presents the main results found after the data analysis, while the discussion is described in Section 4. The future steps and main conclusions are summarized in Section 5.

## 2. Materials and Methods

### 2.1. Participants

From April 2018 to May 2018, 41 relapsing–remitting MS patients were recruited from the MS Clinics at the University Hospital “12 de Octubre” (Madrid, Spain). Patients had clinically confirmed MS according to 2017 revised McDonald criteria [17] and were stable (no relapses) at the study time. Four neurologists (VFS, ADS, SMG, and JBL) with MS expertise conducted the clinical examination using the EDSS to rate the severity of the disease (range = 0–10), [13]. Exclusion criteria were as follows: institutionalized at the observation time, ophthalmological diseases that affected visual acuity despite lens correction, major serious acute or chronic comorbidities, and neurological (other than MS) or psychiatric diseases, including dementia or major depressive disorder.

Forty-three age-matched healthy controls were recruited from the relatives or acquaintances of the study researchers [18]. None of the controls were related to any of the MS patients in the study. In addition, none of them had any known neurological or psychiatric conditions, and no one had a personal history of ophthalmological diseases that, despite lens correction, compromised visual acuity.

In Table 1, the most relevant demographic data of the healthy controls and the MS patients are presented.

In order to guarantee the reproducibility of the data, we used an available and certified gaze-tracker, [19]. The ethical standards committees on human testing at the University Hospital “12 de Octubre” approved all procedures (Madrid). All participants were given oral and written information about the study and signed an informed consent document.

### 2.2. Description of the Eye Movement Tests

A saccadic movement, also known as a “saccade,” is a rapid, jerky eye movement that directs the gaze to a new location and redistributes the region of high visual acuity centered on the fovea [20]. Saccades are voluntary movements but are generally produced with highly automated routines. This section describes the materials used and the method developed to assess the eye movements under binocular vision but with a monocular measurement on the dominant eye. For further information on the neurological aspects of eye movements, the reader is referred to [21].

All participants conducted the visually guided saccades, antisaccades, and smooth pursuit tests in horizontal and vertical axes. The stimulus was a green dot with a 2 cm diameter, and the background color was black. Vertical and horizontal eye movements were recorded under *gap* representation of the visual stimulus.

The gaze tracker consists of a conventional screen where visual stimuli are deployed according to the test (see next sections). The screen is located at 60 cm from a chin rest used to avoid head movements. The eye movement test lasts approximately 10 min. In order to guarantee the accuracy of the measurement, a new calibration was completed every two tests.

#### 2.2.1. Visual Guided Saccade Test

In Figure 1, a scheme of a visually guided saccade is presented. Although this representation is on the horizontal axis, the same values are computed vertically.

The stimulus appears in the center of the screen during 1500 ms and randomly jumps to the right and left (horizontal test) or up and down (vertical test). The instruction to perform this test was “Look at the green dot”. In Table 2, the saccade test is described.

**Definition** **1.**
*Centrifugal latency is the time (▵t(t)) elapsed between the change of the stimulus ($t_{stimulus}$) and the first eye movements performed by the volunteer in response to the stimulus change ($t_{gaze}$). In order to detect this change, the velocity of the movement is computed. We computed it by (Equation 1). In the measure of latency variable (ms), the express saccades (saccadic movement with latency between 80 and 130 ms) were discarded.*

(1)
$$▵t\left(t\right)=t_{gaze}-t_{stimulus}$$



**Definition** **2.**
*Centripetal latency is the time ($▵t_{c}$) elapsed between the change of the stimulus ($t_{stcentre}$) toward the screen center and the first eye movements performed by the volunteer in response to the stimulus change ($t_{gaze}$). This latency is computed by (Equation 2).*

(2)
$$▵t_{c}\left(t\right)=t_{gaze}-t_{stcentre}$$


*According to [18,22,23,24], $▵t_{c}\left(t\right)<▵t\left(t\right)$ in healthy volunteers.*


**Definition** **3.**
*The gain (Equation 3) of the saccade is computed by the ratio between stimulus amplitude ($Amp_{stimulus}$) and gaze amplitude ($Amp_{gaze}$).*

(3)
$$Gain=\frac{Amp_{gaze}}{Amp_{stimulus}}$$


*In previous work, [18], the authors had demonstrated that the accuracy of the saccadic movement is a significant value to evaluate brain aging.*


#### 2.2.2. Antisaccade Test

Figure 2 describes the main parameters measured in the antisaccade test, in which the volunteer must perform a movement in the opposite direction of the stimulus. Table 3 summarizes the main parameters of the test. The instruction to perform this test was: “Look toward the opposite direction to the green dot”.

An antisaccade is correct if the volunteer performs a saccade movement opposite to the stimulus. In order to measure the latency of the antisaccade, we used (Equation 1).

When the subject performs a movement in the direction of the stimulus and then corrects the gaze to the opposite side, this saccade is called “reflexive”. The latency and the duration of the reflexive saccade are indicative of the reaction time of the neural inhibitory pathway [25].

If the reflexive saccade’s duration and the eye movement’s amplitude are higher than 0.5 of the stimulus’s amplitude and duration, then it is counted as a reflexive saccade. In another case, it is a non-identified eye movement. The value of 0.5 was settled practically.

When the subject performs a movement in the direction of the stimulus and does not correct the gaze to the opposite side, this saccade is classified as an error. When the subject performs a movement that does not follow a clear paradigm (prosaccade, reflexive, express, antisaccade), we classified this as an *erratic* antisaccade.

Other variables such as the error rate, corrected saccades, successful, anticipatory, and erratic antisaccades are measured. Figure 3 shows a flow chart describing how we compute these values. If the gaze and stimulus direction are the same, it is a reflexive saccade, and the latency and duration are computed. In the case of a reflexive saccade, we compute two parameters:

**Definition** **4.**
*The Latency of Reflexive Saccade (ms) ($▵t_{ref}$) is the time elapsed from the detection of the eye moving toward the opposite side ($t_{gazeA}$) and the time of the stimulus change ($t_{stimulus}$). It is computed by (Equation 4). See Figure 2.*

(4)
$$▵t_{ref}\left(t\right)=t_{gazeA}-t_{stimulus}$$



**Definition** **5.**
*The duration of reflexive saccades, $\nabla t_{ref}$, (ms), is computed by (Equation 5), and it is the difference between the time at which the subject fixes the gaze on the stimulus ($t_{iniFIX}$) and the time in which the gaze starts the movement to the opposite direction ($t_{endFIX}$):*

(5)
$$\nabla t_{ref}\left(t\right)=t_{endFIX}-t_{iniFIX}$$



In the case the gaze and stimulus direction are opposite, it is a correct antisaccade movement, and latency, dysmetria, and gain are computed using (Equation 1)–(Equation 3), and this eye movement is counted as a success antisaccade. In Figure 3, a flow chart to compute the successful antisaccade eye movement is presented.

#### 2.2.3. Smooth Pursuit Test

Figure 4 describes the smooth pursuit test paradigm. The volunteer must perform a movement following the stimulus. Table 4 summarizes the main parameters of the test. The instruction to perform this test was “Follow the green dot.”

To correctly understand the smooth pursuit test, it is essential to take into account the following concepts: *Catch-up*, *Back-up* and *Square wave jerks.*

**Definition** **6.**
*Catch-up is a saccadic movement in the direction of the smooth pursuit due to the lack of capacity to keep up with the velocity of the dot. The volunteer tries to catch up with the stimulus with a speed saccade.*


**Definition** **7.**
*Back-up is a saccadic movement in the opposite direction of the smooth pursuit because the participant’s gaze is advanced in comparison with the stimulus.*


**Definition** **8.***Square wave jerks are defined as two saccades with an amplitude between 0.2 and 5 grades and in opposites directions. The differences between the directions must be within the interval* 90° <θ< 270°.

**Definition** **9.**
*Gain in smooth pursuit test ($Gain_{sp}$) is a parameter that measures the capability of doing the test. According to [26], we computed it by *(Equation 6)*. If $Gain_{sp}>1$, the eye movement is faster than stimulus movement. If $Gain_{sp}<1$, then the eye movement is slowest.*

(6)
$$Gain_{sp}=\frac{1}{n}\sum_{i=1}^{n}\frac{Vel_{gaze_{i}}}{Vel_{stimulus_{i}}}$$

*where n is the total number of frames, while i is the current frame where the velocities are computed.*


**Definition** **10.**
*Time of pursuit is a parameter that measure the effective time that the subject is following the stimulus without intrusive saccades [26].*


## 3. Results

### 3.1. Data Analysis Methods

To compare eye movement parameters between healthy controls and MS patients, we adjusted multiple linear regression models (Equation 7) that predicted eye movement parameters ($Y_{i}$) according to the presence of disease ($X_{disease}$) controlling the effect of age ($X_{age}$). Coefficient $\beta_{1i}$ represents the change in the eye movement parameter *i* attributable to MS ($X_{disease}=1$) in comparison to control group ($X_{disease}=0$) controlled by the effect of increasing one year of age ($\beta_{2i}$). $\beta_{0i}$ represents the intercept of the multivariate regression, and ϵ is the error.
(7)$$Y_{i}=\beta_{0i}+\beta_{1i}*X_{disease}+\beta_{2i}*X_{age}+\epsilon$$

Moreover, we aim to explore the relationships between the selected eye movement parameters and disease duration ($X_{duration}$), controlling the effect of age using multiple linear regression (Equation 8). Coefficients $\beta_{1i}$ and $\beta_{2i}$ represent the change in the eye movement parameter *i* of having an additional year of MS duration.
(8)$$Y_{i}=\beta_{0i}+\beta_{1i}*X_{duration}+\beta_{2i}*X_{age}+\epsilon$$

Finally, the partial Spearman’s correlation for EDSS and the selected eye movement parameters controlling the age effect were computed. We used percentile bootstrap for 95% confidence interval calculation of the correlation coefficient.

In order to represent how the different abnormalities seen in the eye movement parameters correlate with each other in MS, we have used a heat map. We used hierarchical clustering to classify both patients and parameters. In both cases, we used the average as the link criteria. In the case of patients, we have used Gower’s distance with the aim of decreasing the impact of different distributed parameters. Furthermore, we have used a parameter based on Spearman’s correlation coefficient: see Equation (Equation 9) in which we defined the distance between variable *i* and *j*.
(9)$$d_{ij}=1-\left|\rho_{ij}\right|$$

A color scale was used based on Z-score standardization (according to the healthy subjects’ mean and standard deviation) to represent each patient’s parameters. Equation (Equation 10) shows the Z-score for the patient *p* in variable *k*. Since some parameters do not have a normal distribution, we first transform the data of the reference subjects in a normal distribution using a suite of normalizing transformations [27].
(10)$$Z_{k_{p}}=\left(k_{p}-mean\left(K_{healthy}\right)\right)/sd\left(K_{healthy}\right)$$

### 3.2. Comparison between Healthy Controls and MS Patients

Figure 5, Figure 6 and Figure 7 show the distribution of the selected parameters described in Section 2 for both groups. These parameters were selected based on the analysis presented in Figure 8. Although there was a tendency for increased latencies in the case of visually guided saccades (both centripetal and centrifugal saccades), a subgroup of MS patients had the latency (horizontal and vertical) increased compared with controls. The accuracy of the saccadic movement was worse in MS patients than in controls. While the control group had a gain near 1, there were MS patients who had hypometric and hypermetric saccades. According to the distribution of this parameter in Figure 5, MS patients tended to have hypometric vertical saccades. Figure 6 shows the antisaccade test distribution. The vertical and horizontal latencies of reflexive saccades were increased in MS patients, which was mainly due to the increased duration of reflexive saccades. There was also a tendency for decreased percentages of correct saccades with more anticipatory saccades and a relative increase in the return latency. Parameters in the smooth pursuit test (see Figure 7) were those who showed more abnormalities. On the horizontal axis, MS patients showed more backup saccades; on the vertical axis, they had decreased gain, more catch-up saccades, and decreased pursuit time. It is important to highlight that these changes were controlled for the age effect.

When comparing the *p*-values given in Figure 8 and Figure 9, MS patients showed more alteration in the vertical direction. Moreover, there was no significant correlation between the EDSS total score, eye movement parameters, and disease duration. The gain at horizontal visually guides saccades showed a significant association with disease duration. The percentage of errors at horizontal and vertical saccades had a statistically significant coefficient, but bootstrap intervals were negatives. Patients with higher EDSS total scores exhibited a significantly increased return latency in horizontal antisaccades with less corrected saccades and a tendency to have less successful antisaccades. Moreover, an increased latency in the horizontal movement was observed in the smooth pursuit.

Finally, we studied the multivariate distribution of the abnormalities found in the group of patients with MS. This result is represented in Figure 10. Significantly, there was no identifiable pattern in the patients, and eye movement parameters show poor correlations between them.

## 4. Discussion

Eye movement parameters are considered precise, low-cost, functionally related biomarkers for clinical monitoring MS [28]. In our study, we have demonstrated the plausibility of a quasi-automatic system for measuring eye movement performance in a relatively large group of MS patients. The oculomotor system seems particularly useful in the study of motor control and cognitive function due to several reasons: (a) well-known neural basis in primates, (b) well-known neuroimaging correlates, and (c) easy and robust parameters to assess each movement paradigm [25]. The development of accessible, simple, functional biomarkers is particularly relevant in MS, in which biomarkers are mainly structural; available functional biomarkers only monitor walking or dexterity performance and show a limited sensitivity to change [29]. In addition, cognitive status evaluation remains challenging in routine clinical settings and research [30].

We have used several paradigms to assess ocular motor performance. Visually guided saccades are the simplest ones. Although we did not find a significant difference in most parameters, we detected a tendency in a subgroup of patients with values beyond the limits of normality, which were defined by the healthy control group. Increased latencies may reflect a prolonged processing time related to attentional problems [31,32,33]. In addition, some patients had inaccurate saccadic movements, which may indicate problems in motor control due to problems in the cerebellum and cerebellar-dependent circuits [34].

In contrast to visually guided saccades, which are generated in response to external cues and require a more simple sensorimotor implementation, antisaccades are a volitional paradigm in which movement is far more dependent on cognitively complex responses and requires the recruitment of higher-order control processes (inhibition, spatial memory, and analysis of contextual cues) [21,25,35]. We found an increased time to correction in reflexive antisaccades and in the duration of reflexive saccades, which may also indicate damage in frontal circuits depending on frontal eye field [36]. A wide distribution of values exists for these parameters in healthy subjects, which hampers interpretation in patients with MS. Anticipatory saccades tend to be increased in MS patients. Again, there is a subgroup of patients with a clear increment, which may reflect a problem in networks involving basal ganglia [37,38,39,40]. The mean percentages of successful or corrected saccades in MS patients were similar to healthy controls.

In visually guided saccades and antisaccades, the centripetal saccades’ latency tended to increase compared to healthy subjects. These saccades have an exogenous component but are facilitated by an endogenous component [41]. The ability of MS patients to include this previous information and prepare the response field of saccadic movement generators seems to be impaired, which fits with the relationship between cognitive problems and eye motor abnormalities in MS patients.

Significantly, we found important abnormalities in the values of smooth pursuit. Latency showed similar values, indicating that the initiation of this type of eye movement was relatively preserved. In contrast to latency, which showed similar values to healthy control, the parameters related to the quality of motor control during the pursuit were or tended to be altered except for gain. Abnormalities in smooth pursuit occur in various conditions [42,43,44]. This reflects an impairment of a complex system involving the middle temporal cortical visual area, dorsolateral pontine nuclei, and cerebellum [45].

Interestingly, we could not find a clear common pattern in the eye motor abnormalities. Eye movement parameters were also poorly correlated between them. We interpret this as a consequence of the patchy, heterogeneous distribution of white matter involvement in MS that provokes multiple combinations of impairment at different points in the different networks involved in eye motor control [46]. This also has a direct consequence in assessing eye movement disorders in patients with MS, as we will need several tasks to cover the different functional domains.

Although some previous work in the literature reports the alteration in eye movements in MS disease, most of them lack technical details related to the measurement technique and how the parameters were computed. Even so, there is a consensus on the impairment of eye movements in these patients. In [47], a cohort of 163 MS patients with MRI was recruited. The authors concluded that ocular movement alterations are frequent in patients with stable (no relapses) MS. At least 68.1% of patients showed at least one eye movement abnormality. The most frequent eye movement alterations were impaired smooth pursuit 42.3%, saccadic dysmetria 41.7%, and latency in saccades 14.7%. The results of recording eye movement to 50 MS patients with nystagmus were presented in [48]. The authors used EyeLink at 250 Hz. The main finding was that the pendular nystagmus seen in MS patients was mainly disconjugated, and that this disconjugacy was related to the amplitude and not the frequency of nystagmus. Moreover, other oculomotor abnormalities were observed in most MS patients, although their presence was unrelated to the MS severity or stage.

Impaired saccadic eye movements were related to the functional connectivity of the oculomotor network and clinical performance in MS in [49]. The authors measured the eye movements using Eyelink 1000 Plus. Off-line analysis was performed in Matlab, and the results were the average value of the right and left eyes. The main finding was that pro-saccadic latency was strongly associated with disability scores and cognitive dysfunction.

In [50], thirty-three MS patients and twenty-five matched healthy control subjects were included in a study of eye movements. The Symbol Digit Modalities Test (SDMT) was implemented for both groups. That study aimed to determine the association between eye movements and performance on the SDMT. The authors concluded that cognitive-mediated eye movements help elucidate the processing speed challenges confronted by MS patients, which can potentially help inform new cognitive rehabilitation strategies.

Eye tracking has been used to screen for abnormal visuospatial behavior in MS [51]. Ref. [52] designed an ocular working memory task in which participants were instructed to recall the positions of numerical stimuli on a screen while being recorded by an eye tracker. The results showed higher error rates in the MS group, with more working memory errors made by patients farther along in the disease course, which was in line with previous studies [52,53,54]. In [55], the authors conducted a battery of unpredictable, predictable, and endogenously cued visually guided saccade tasks on MS patients. Participants were asked to look directly at the center of a green target cross and then pursue the target as it moved horizontally across the screen while ignoring a visual distractor. Saccadic latency and absolute position error were measured. Compared with a healthy group, MS patients exhibited increased saccadic latency [6,7,51,56], worse fixation, and more prosaccade errors. These results demonstrated eye-tracking’s diagnostic and prognostic potential to assess cognitive function and disease severity in MS [7].

Abnormalities in the vertical plane were more common than in the horizontal one. This is a common observation in other disorders and highlights the relevance of exploring the vertical axis (or others). The reader is referred to [57] for a comprehensive review.

We are aware that this study has several limitations. The most important is the sample size. The high variability in MS eye motor performance and low sample size increase the uncertainty regarding the effect size of the differences or relationships. In particular, negative results are complicated to interpret. We, however, considered that the sample size was enough to detect robust differences between patients and controls in cognitive domains that are affected by MS [8,9]. Notwithstanding, these findings should be taken cautiously until they are replicated in larger, independent data sets. Second, we recruited a group of MS patients who were seen in a clinic; as a result, our results may not apply to MS patients in the general population. Therefore, we acknowledge the need to replicate these findings in an unselected patient population-based survey. Another important idea to cover in further studies is the joint evaluation of outcomes coming from the afferent (e.g., low-contrast vision quality or optical coherence tomography) and the efferent visual system (eye motor abnormalities. Finally, the participants did not complete a neuropsychological battery that would have provided additional insight into the origins of differences in eye movements between MS patients and healthy controls.

## 5. Conclusions

This article presents a systematic procedure to assess the cognitive dysfunction in a group or patient with relapsing–remitting multiple sclerosis. We have proposed to use an alternative technique: that is, to measure eye movements. Although we implemented a specific test battery under a systematic procedure, we demonstrated that patients had alterations in eye movement behavior compared with healthy controls. However, we were not able to define a common pattern among patients. A subgroup of patients showed similar behavior. However, we could not define a pattern even for this subgroup. This finding is coherent with the disease characterization: the disease of a thousand faces. Further studies are therefore required.

## Figures and Tables

**Figure 1 sensors-22-08220-f001:**
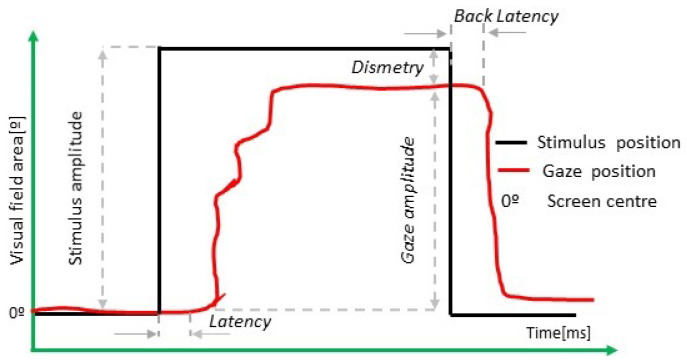
Main parameters measured in visually guided saccade test.

**Figure 2 sensors-22-08220-f002:**
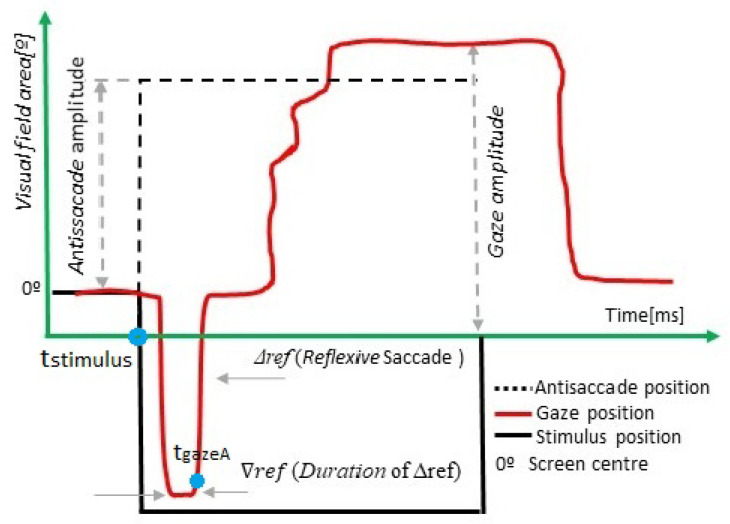
Antisaccade test.

**Figure 3 sensors-22-08220-f003:**
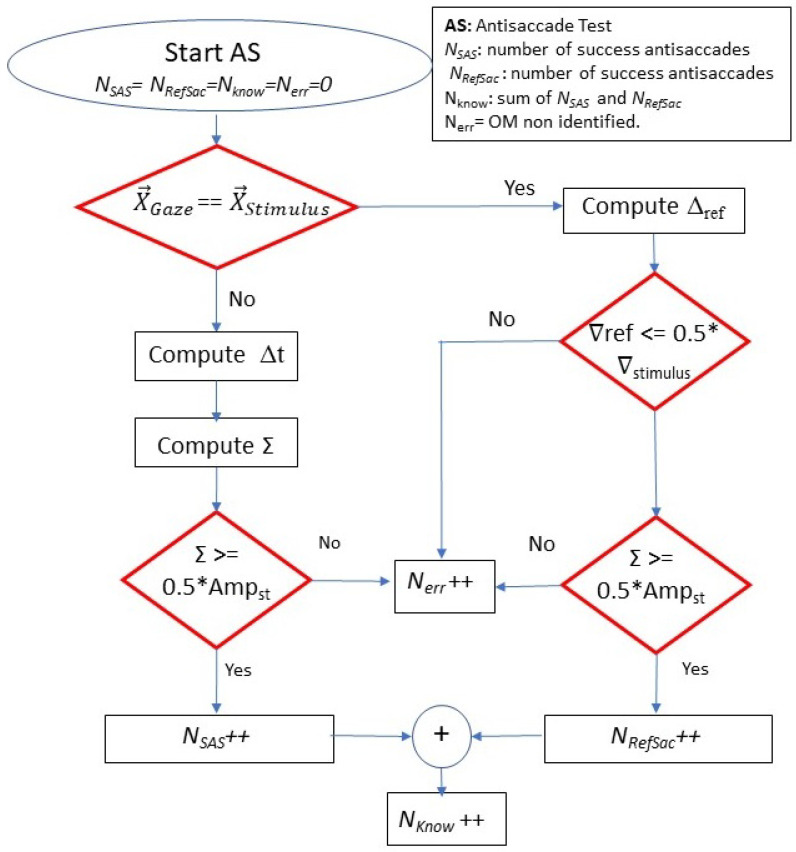
Flow chart to compute success rate in antisaccade test.

**Figure 4 sensors-22-08220-f004:**
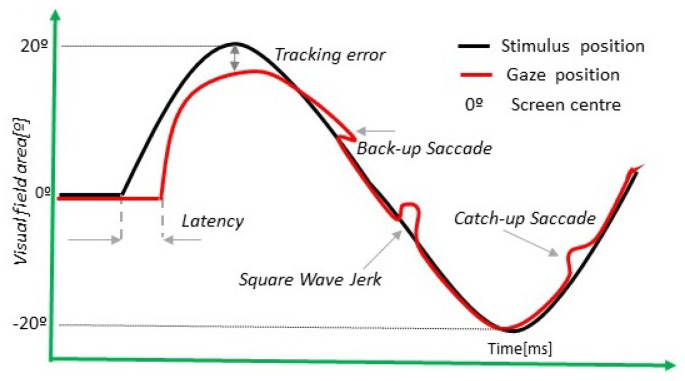
Smooth pursuit test.

**Figure 5 sensors-22-08220-f005:**
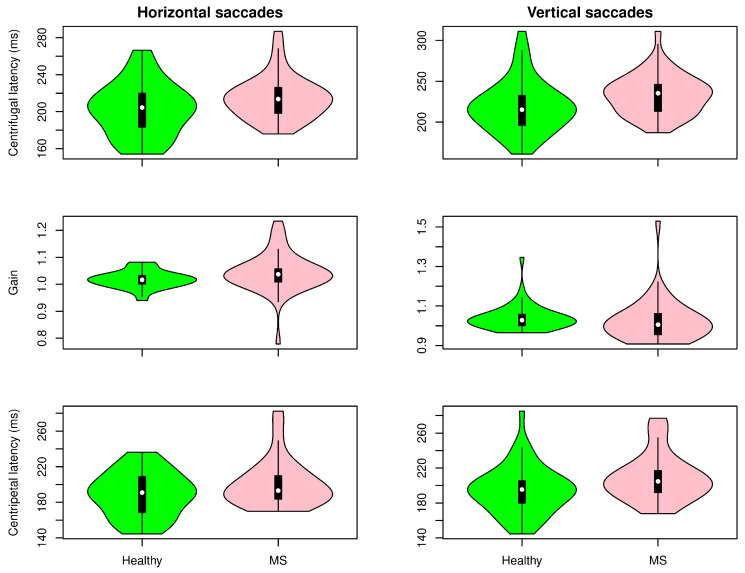
Visually guided saccade test results. Distribution of selected parameters in horizontal (**left**) and vertical (**right**) saccades for healthy volunteers and multiple sclerosis patients.

**Figure 6 sensors-22-08220-f006:**
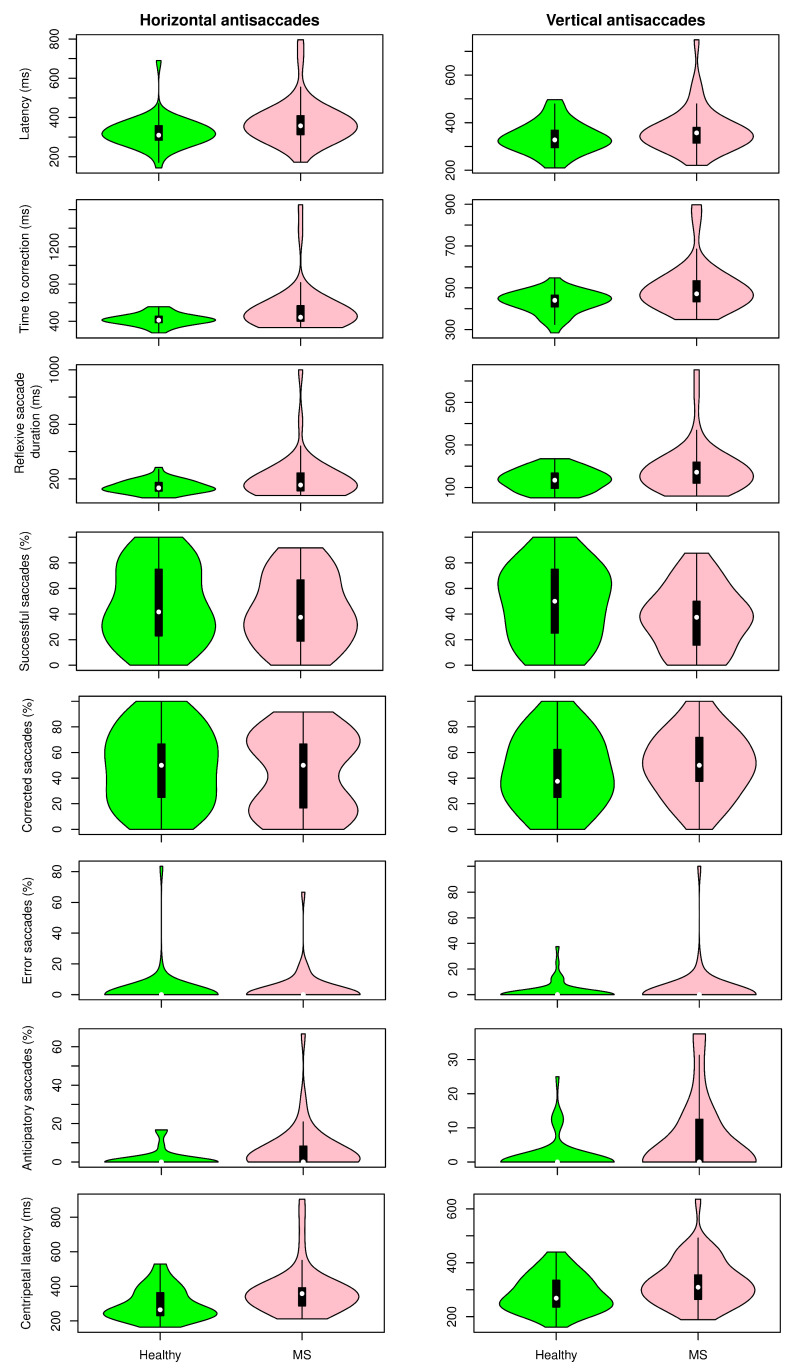
Results for the antisaccade task. Distribution of selected parameters in healthy volunteers and multiple sclerosis patients.

**Figure 7 sensors-22-08220-f007:**
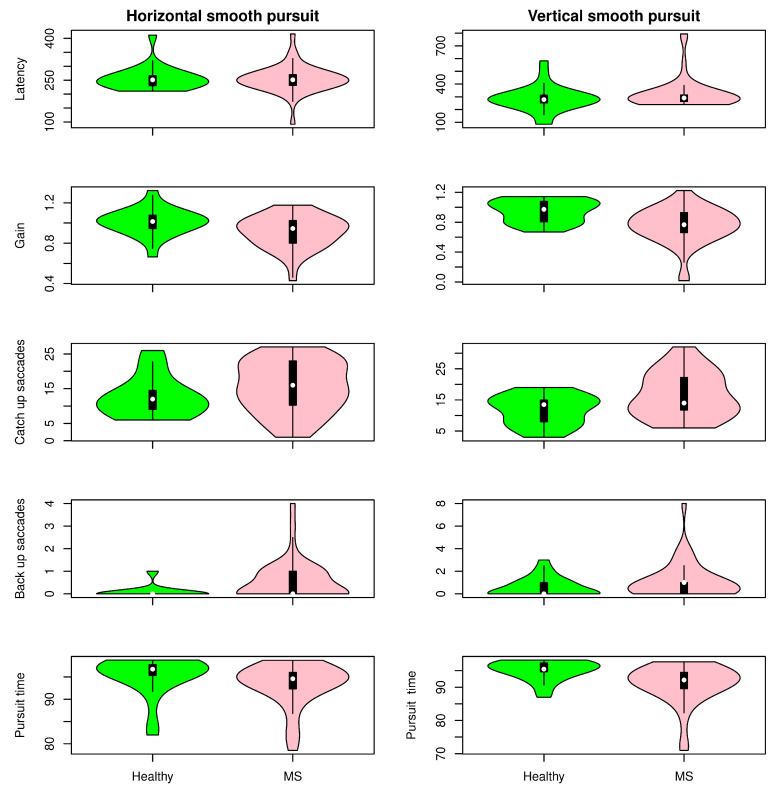
Results for the smooth pursuit task. Distribution of selected parameters in healthy volunteers and multiple sclerosis patients.

**Figure 8 sensors-22-08220-f008:**
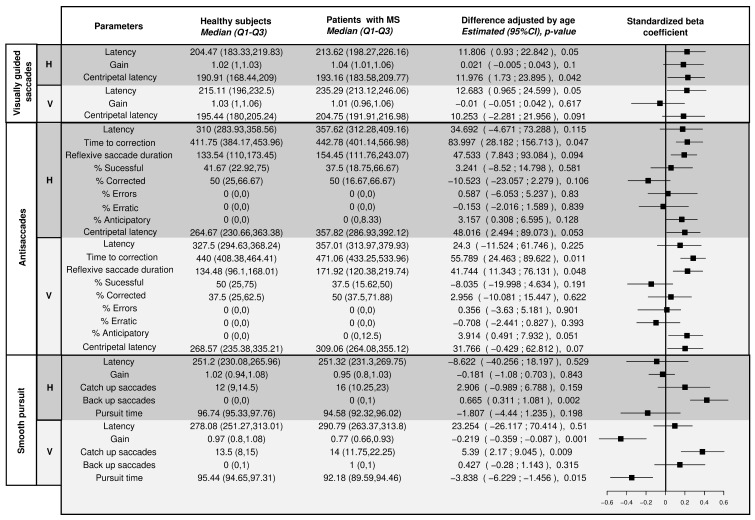
Differences between healthy subjects and multiple sclerosis patients. Significance *p*-values < 0.05.

**Figure 9 sensors-22-08220-f009:**
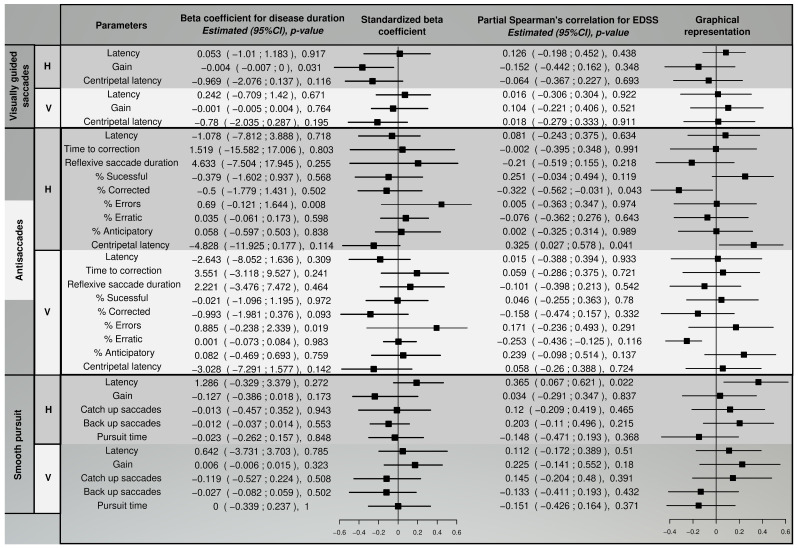
Correlation results between EDSS total score, disease duration, and eye movement parameters. Significance *p*-value < 0.05.

**Figure 10 sensors-22-08220-f010:**
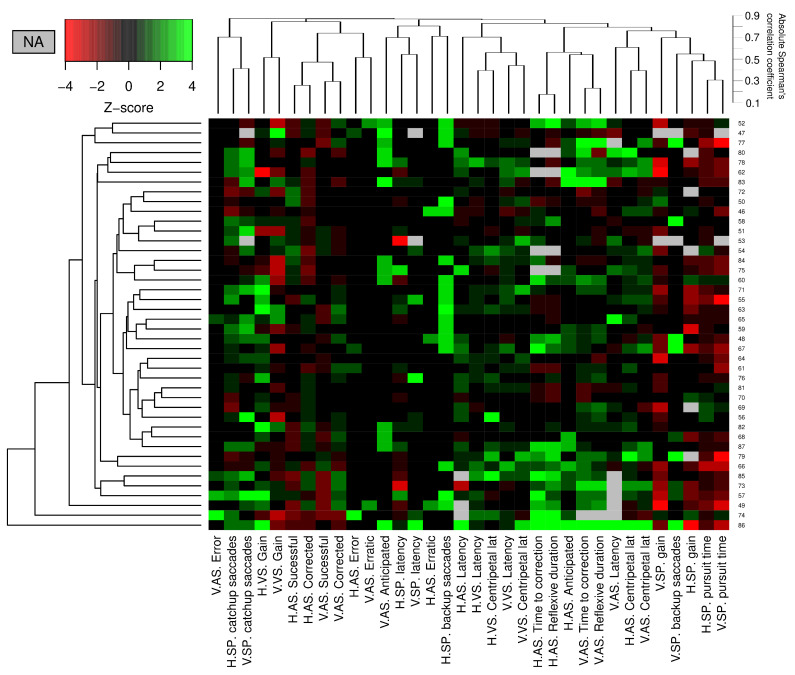
Heat map that shows the distribution of the eye motor abnormalities in patients with MS. Every value for each included parameter (column) is shown for each patient (rows) according to its Z-score value calculated with data from healthy reference subjects. Patients are ordered according to a hierarchical clustering based on Gower’s distance, whose result is represented in the dendrogram at the left part of the figure. Parameters are ordered according to a hierarchical clustering based on Spearman’s rho correlation coefficient, whose result is shown in the upper dendrogram. The average Spearman’s rho value is indicated at the right of the dendrogram. Notice the lack of clear patterns.

**Table 1 sensors-22-08220-t001:** Demographic Data. Healthy Controls and Multiple Sclerosis Patients.

Group	Sample Size	Age	EDSS	Age at Diagnosis	Disease Duration
Controls	43	33.58± 9.47	―	―	―
Patients	41	38.69± 7.50	0.76± 1.13	30.45± 8.33	9.43± 7.13

**Table 2 sensors-22-08220-t002:** Default parameters for visual guide saccades test.

Mode	Visual Field (°)	Duration (s)	Repetitions
Horizontal	5, 10, 20	36	22
Vertical	5, 12	24	12

**Table 3 sensors-22-08220-t003:** Default parameters for antisaccades test.

Mode	Visual Field (°)	Duration (s)	Repetitions
Horizontal	5, 10, 20	36	22
Vertical	5, 12	24	12

**Table 4 sensors-22-08220-t004:** Default parameters for smooth pursuit test.

Mode	Duration (s)	Repetitions
Horizontal	8	3
Vertical	8	3

## Data Availability

Data are available for researchers upon reasonable request.

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
