# Peer review of "Toward an Automatic Assessment of Cognitive Dysfunction in Relapsing–Remitting Multiple Sclerosis Patients Using Eye Movement Analysis"

_sensors, 2022, doi:10.3390/s22218220_

Round 1

Reviewer 1 Report

The work of García Cena1 et al. analyzed the association between eye movement and cognitive dysfunction in relapsing-remitting multiple sclerosis patients based on a purposeful sampling of 84 patients. The study demonstrated that there is no clear pattern in the eye motor abnormalities. And no correlation was found between eye movement variables and EDSS or disease duration. The paper is well-written but some major corrections need to be made.

Participants
The author recruited 41 relapsing-remitting MS patients and 43 healthy controls. The control groups were matched with age. However, a simple calculation based on an independent t-test reveals these two groups’ age are significantly different. As a result, age could still be interfering with the detection of eye movement related to MS after ineffective matching.

The most serious issue is the sample size, although the author has revealed the insufficient sample size issue in the Discussion section, it is expected for the author the address more concerns related to the sample size. Since this study reports a negative result, whether the obtained number of cases achieved sufficient power would be crucial before concluding. The insignificant correlation found between eye movement and EDSS could be due to the lack of statistical power.

Statistical analysis
Besides age, other covariates might potentially affect the eye movement in MS patients that the author could consider including in the regression model, such as gender, other eye disease history, length of MS disease history, and many others. 

Author Response

Please find details in the attached document

Reviewer 2 Report

This article about the systematic procedure to assess the cognitive dysfunction in a  group or patient with relapsing-remitting multiple sclerosis. overall paper looks good and sound. need to address the below queries

-abstract can be enhance well in terms of all. remove the words like introduction, results, conclusion etc in this

- there is no single reference in the lst few paragraphs in introduction section. check them. if required cite relevant papers

-related works section can be added to discuss the recent works happened in this field so users can understand before going to your work also. can be refer, hybrid genetic algorithm and a fuzzy logic classifier for heart disease diagnosis

- how did you collect this data to analyse and howmany patients data gathered. or any online source used

- what are the evaluation metrics are considered to evaluate this work and compared with other existing works

-Fig. 2 describes the main parameters measured in the antisaccade test, in which the 129 volunteer must perform a movement in the opposite direction of the stimulus. Please explain more about this image and its usage

-what is success rate achieved in antisaccade test. whats the position when you compared with other existing methods

-what is the difference between back up saccade and catch up saccade which mentioned in the figure 4

- please explain all attributes used in the equations clearly. mainly in equations 7 and 8

Author Response

Please find details in the attached document.
